# ASPN Is a Potential Biomarker and Associated with Immune Infiltration in Endometriosis

**DOI:** 10.3390/genes13081352

**Published:** 2022-07-28

**Authors:** Li Wang, Jing Sun

**Affiliations:** Department of Gynecology, Shanghai First Maternity and Infant Hospital, School of Medicine, Tongji University, Shanghai 200092, China; wlkykt2016@126.com

**Keywords:** ASPN, endometriosis, immune infiltration, GEO, biomarker

## Abstract

**Objective:** Endometriosis is a benign gynecological disease characterized by distant metastasis. Previous studies have discovered abnormal numbers and function of immune cells in endometriotic lesions. We aimed to find potential biomarkers of endometriosis and to explore the relationship between ASPN and the immune microenvironment of endometriosis. **Methods:** We obtained the GSE141549 and GSE7305 datasets containing endometriosis and normal endometrial samples from the Gene Expression Omnibus database (GEO). In the GSE141549 dataset, differentially expressed genes (DEGs) were found. The Least Absolute Shrinkage and Selection Operator (Lasso) regression and generalized linear models (GLMs) were used to screen new biomarkers. The expression levels and diagnostic utility of biomarkers were assessed in GSE7305, and biomarker expression levels were further validated using qRT-PCR and western blot. We identified DEGs between high and low expression groups of key biomarkers. Enrichment analysis was carried out to discover the target gene’s biological function. We analyzed the relationship between key biomarker expression and patient clinical features. Finally, the immune cells that infiltrate endometriosis were assessed using the Microenvironment Cell Population-Counter (MCP-counter), and the correlation of biomarker expression with immune cell infiltration and immune checkpoints genes was studied. **Results:** There were a total of 38 DEGs discovered. Two machine learning techniques were used to identify 10 genes. Six biomarkers (SCG2, ASPN, SLIT2, GEM, EGR1, and FOS) had good diagnostic efficiency (AUC > 0.7) by internal and external validation. We excluded previously reported related genes (SLIT2, EGR1, and FOS). ASPN was the most significantly differentially expressed biomarker between normal and ectopic endometrial tissues, as verified by qPCR. The western blot assay revealed a significant upregulation of ASPN expression in endometriotic tissues. The investigation for DEGs in the ASPN high- and low-expression groups revealed that the DEGs were particularly enriched in extracellular matrix tissue, vascular smooth muscle contraction, cytokine interactions, the calcium signaling pathway, and the chemokine signaling pathway. High ASPN expression was related to r-AFS stage (*p* = 0.006), age (*p* = 0.03), and lesion location (*p* < 0.001). Univariate and multivariate logistic regression analysis showed that ASPN expression was an independent influencing factor in patients with endometriosis. Immune cell infiltration analysis revealed a significant increase in T-cell, B-cell, and fibroblast infiltration in endometriosis lesions; cytotoxic lymphocyte, NK-cell, and endothelial cell infiltration were reduced. Additionally, the percentage of T cells, B cells, fibroblasts, and endothelial cells was favorably connected with ASPN expression, while the percentage of cytotoxic lymphocytes and NK cells was negatively correlated. Immune checkpoint gene (CTLA4, LAG3, CD27, CD40, and ICOS) expression and ASPN expression were positively associated. **Conclusions:** Increased expression of ASPN is associated with immune infiltration in endometriosis, and ASPN can be used as a diagnostic biomarker as well as a potential immunotherapeutic target in endometriosis.

## 1. Introduction

Endometriosis is defined as the pathological changes caused by endometrial tissue growth and infiltration outside the uterine cavity. The disease is estrogen-dependent, with clinical symptoms such as menstruation, infertility, etc. In recent years, the incidence has increased significantly, reaching 10–15% and as high as 25–40% in infertile women [1,2]. The disease is clinically classified into peritoneal endometriosis, ovarian endometriosis, and deep infiltrative endometriosis according to the location of the ectopic lesion in the pelvis. The main pathological changes of the disease are bleeding of the ectopic endometrium, fibrosis of the surrounding tissue, and the formation of ectopic nodules.

In recent years, treatment techniques related to endometriosis have been advancing. Still, the effectiveness of existing standard treatments is limited due to the aggressive nature of endometriosis and the high postoperative recurrence rate. Laparoscopy is the gold standard for the diagnosis of endometriosis, but the invasive nature of the technique limits its widespread clinical use [3]. Ultrasound and MRI are commonly used to diagnose and evaluate endometriosis conditions. Still, the sensitivity of imaging varies depending on the location of the endometriosis lesion, while the detection rate of imaging for early endometriosis lesions is extremely low [4,5]. Many biochemical markers are widely used in clinical diagnosis, including CA125, HE4, etc. However, these methods are insufficient for the early diagnosis of endometriosis due to sensitivity and specificity limitations. Therefore, it is essential to investigate more potential molecular targets for their diagnosis, treatment, and prognostic assessment.

Although the exact pathogenesis remains unclear, current research suggests that endometriosis is a disease caused by immune-related mechanisms, and the degree of immune cell infiltration is a significant factor affecting the prognosis of the condition [6,7]. Research by Cui et al. has shown that the changes in macrophage phenotype promote endometriosis by maintaining the growth of ectopic cells [8]. It has been observed that endometriosis patients have a considerable increase in Th2 expression in their peritoneal fluid [9]. Furthermore, there have been reports that the cytotoxic function of NK cells is weakened, which mediates immune escape and causes endometrial cell adhesion and migration [10]. Therefore, evaluating differences in the composition of immune infiltrating cells in endometriosis is of great value to elucidate their molecular mechanisms and identify molecular markers associated with immune infiltration.

Microarray technology has been increasingly popular for identifying potential biomarkers of complicated disorders [11,12]. However, to date, there are fewer bioinformatics analyses of endometriosis. This research aimed to identify immune-related endometriosis biomarkers that could increase the accuracy of an endometriosis diagnosis. In our study, the GSE141549 dataset was downloaded from GEO to identify DEGs between endometriosis and controls. Lasso regression and generalized linear model algorithms are used to screen diagnostic biomarkers for endometriosis. We confirmed the diagnostic efficacy of biomarkers in the GSE7305 dataset. Further validation of the expression levels of the biomarkers was carried out using qRT-PCR and western blot tests. The study found that ASPN may be a new biomarker for endometriosis. Univariate and multifactorial logistic regression models were further constructed to analyze the correlation between ASPN expression and the clinical characteristics of patients with endometriosis. Finally, the immune cell composition between endometriosis and normal tissue was quantified using the MCP-counter. By studying the relationship between ASPN, immune infiltration, and immune checkpoint genes, we can better understand the molecular immune mechanisms involved in the pathogenesis of endometriosis.

## 2. Materials and Methods

### 2.1. Patients and Tissue Samples

In our study, twelve normal endometrial specimens and twelve samples of ectopic lesions from patients with endometriosis were collected from patients who underwent surgical treatment at the Shanghai First Maternal and Infant Hospital affiliated with Tongji University. Two highly qualified pathologists independently diagnosed these samples. Before use, samples were stored in liquid nitrogen tanks, and the Ethics Committee approved this study at Shanghai First Maternal and Infant Hospital (KS21198). All patients provided their voluntary informed consent.

### 2.2. Data Download

This study uses the GSE7305 [13] and GSE141549 [14] datasets from the GEO database [15]. These data were derived from the GPL570 platform (Affymetrix Human Genome U133 Plus 2.0 Array) and the GPL13376 platform (Illumina HumanWG-6 v2.0). GSE7305 contains mRNA information from the endometrium of 10 cases of endometriosis and 10 healthy women. The GSE141549 dataset consisted of 408 samples, of which 335 were endometriosis samples and 66 were healthy control samples; the status of 7 samples was unknown.

### 2.3. Identification of Differentially Expressed Endometriosis Genes

We downloaded the normalized and log-transformed GSE141549 data. Gene expression was analyzed using the limma package [16] and DEGs according to the following criteria: |log fold change (FC)| > 1 and an adjusted *p*-value < 0.05. The ggplot2 package was used to draw the differential genes’ heat and volcano map (https://ggplot2.tidyverse.org, accessed on 20 December 2021) with R software (Version 4.0.2, Robert Gentleman and Ross Ihaka, New Zealand).

### 2.4. Candidate Diagnostic Biomarker Screening

We adopted the Lasso regression and generalized linear model to search for diagnostic indicators for endometriosis. The Lasso regression filters feature variables by constructing penalty functions and compressing coefficients. The “glmnet” package in R was used to screen the genes associated with endometriosis. To evaluate the predictive value of the discovered biomarkers, the candidate genes were further validated on the GSE7305, and their diagnostic significance was assessed using receiver operating characteristic (ROC) curves. After excluding related genes previously reported in the literature, the results were presented in a Venn diagram.

### 2.5. Quantitative Real-Time PCR (qRT-PCR)

We took 50 mg of tissue using Trizol reagent (RK30129#cat; ABclonal, Wuhan, China) to extract total RNA, measure RNA concentration and A260/280, and evaluate RNA quality. RNA was reverse-transcribed using a reverse transcription reagent (RK20428; ABclonal, Wuhan, China) to obtain cDNA, and 1ul of cDNA was taken to prepare the qPCR system. PCR reactions were performed using Genious 2X SYBR GREEN Fast qPCR Mix reagents (RK21206; ABclonal, Wuhan, China) and a qRT-PCR device (QuantStudio5, Thermo Fisher Scientific, Waltham, MA, USA). The relative expression of ASPN mRNA was calculated by the 2^−^^△△^Ct method using the GAPDH gene as an internal reference. The primers were synthesized by Shanghai Sangon Biotechnology Company, ASPN primer sequence, forward: 5′-TGATCTGTTTCCAATGTGTCC-3′; reverse: 5′-ACTGAGGTCAAACCTAAATC-TG-3′. Internal reference GAPDH primer sequence, forward: 5′-CTGACTTCAACAGCGACACC-3′, reverse: 5′-TGCTGTAGCCAAATTCGTTGT-3′.

### 2.6. Western Blotting Experiments

After extraction of total tissue proteins from RIPA lysates, the protein concentrations obtained were determined using the BCA Protein Quantitative Assay Kit. After denaturing the proteins by boiling for 10 min, the protein samples (20 ugs per well) were subsequently separated by electrophoresis on a 10% SDS-PAGE gel and wet-transferred to a PVDF membrane. The membranes were closed in skim milk for 1 h and incubated overnight at 4 °C by adding primary antibody rabbit anti-ASPN antibody (1:2000; ABclonal, A10311); rabbit anti-GAPDH (1:7000; New Cell & Molecular Biotech, Suzhou, Jiangsu Province, China, P04406) served as the internal control. They were then washed three times with TBST solution. The membranes were incubated at room temperature for 1 h by adding a horseradish peroxidase-labeled secondary antibody and washed. Finally, the development was performed in the darkroom, and the grayscale values of the target bands were analyzed after taking photographs.

### 2.7. Functional Enrichment Analysis

The enrichment analysis in this study included Gene Ontology (GO)/Kyoto Encyclopedia of Gene and Genomes (KEGG) analysis and Gene Set Enrichment Analysis (GSEA). GO is a bioinformatics tool used to annotate genes and analyze genetic and biological processes. The KEGG database is widely used for pathway enrichment analysis, which yields critical pathways in the development of endometriosis. GSEA is used to explore if a set of genes may exhibit consistent, substantial differences in two biological states [17]. Taking the ASPN median value as the truncation value, the samples were divided into ASPN high-expression group and low-expression group, and the different genes of the two groups were screened using limma packets for GO/KEGG analysis. The GSEA identified the most important set of functional genes between the ASPN high- and low-expression groups. The ranking of the number of enriched genes, the enrichment score, and the normalized *p*-value were combined to determine if the gene set was significantly enriched.

### 2.8. Immune Infiltration Level Analysis

Based on marker gene sets, MCP-counter is a technique for quantifying the immune cells, fibroblasts, and epithelial cells that infiltrate tumors [18]. We assessed the level of immune cell infiltration in both groups using MCP-counter. Corrplot and ggplot2 packages were used to visualize immune infiltrating cells’ correlation and infiltration differences. The relationship between ASPN expression and immune infiltrating cell subsets, immune cell gene markers, and immune checkpoint genes was examined using Spearman’s rank correlation analysis.

### 2.9. Statistical Analysis

For data analysis, we used SPSS 25.0 software and R software (version 4.0.2), the continuous variables were described by mean ± SD, and a *t*-test was used to compare groups if the data were normal; if the variables did not conform to a positive attitude distribution, a non-parametric rank-sum test was used. The paired x test was used for the count data; a logistic regression analysis model was used to test the significance of relevant clinical indicators. The ability of characteristic genes to differentiate between the endometriosis group and the control group was assessed using the ROC curve. When the *p*-value was less than 0.05, all statistical tests were deemed to be statistically significant.

## 3. Results

### 3.1. Identification of DEGs in Endometriosis

We retrospectively analyzed the data of 335 cases of endometriosis and 66 control samples in the GSE141549 dataset (Table 1). A total of 38 DEGs were identified in GSE141549 using the following criteria: |log fold change log (FC)| > 1 and an adjusted *p*-value of 0.05. These DEGs included 34 up-regulated and four down-regulated genes (Figure 1A,B).

### 3.2. Screening Validation of Diagnostic Markers

Lasso regression is a machine learning algorithm to obtain a more compact model by constructing a penalty function, which has a unique advantage for handling complex covariance data and is now commonly used for variable filtering and model complexity reduction [19,20]. The generalized linear model is an extension of the traditional linear model, an algorithm in which the overall mean is passed through a nonlinear connectivity function to handle and take non-normally distributed data well [21]. To screen out the best biomarkers for endometriosis, we used the Lasso regression algorithm to identify 26 variables as potential diagnostic markers from 38 DEGs (Figure 2A,B). Subsequently, a subset of 10 genes among the 26 characteristic variables was determined using the generalized linear model (Table 2).

The area under the ROC curve (AUC) value was used to explore the sensitivity and specificity of 10 variables for identifying endometriosis. The AUCs of SCG2, EGR1, ASPN, SLIT2, FOS, GEM, and GREM1 were all greater than 0.7 (Figure 3). We used GSE7305 as a validation dataset to assess the diagnostic ability of the biomarkers. Six candidate genes for endometriosis showed a high diagnostic capacity, with an AUC of 1.000 (95% CI 1.000–1.000) for SCG2, an AUC of 0.840 (95% CI 0.658–1.000) for GER1, an AUC of 0.930 (95% CI 0.807–1.000) for ASPN, an AUC of 0.990 (95% CI 0.958–1.000) for SLIT2, an AUC of 0.805 (95% CI 0.607–1.000) for FOS, and an AUC of 0.800 (95% CI 0.591–1.000) for GEM (Figure 4).

Three novel biomarkers (SCG2, ASPN, and GEM) were screened, and the findings are displayed in Figure 5A. The genes (EGR1, SLIT2, and FOS) were rejected because they were previously correlated with the development of endometriosis in the literature. The relative expression of mRNA for ASPN and GEM was found to be higher in endometriosis tissues than in normal controls by qRT-PCR (*p* = 0.004, *p* = 0.039; n = 6 vs. 6). There was no discernible difference between the two groups’ SCG2 mRNA expression (*p* = 0.177; n = 6 vs. 6) (Figure 5B). ASPN, the most differentially expressed gene, was selected as the target gene for our study, and the sample was further expanded to verify its transcriptional and protein-level expression. mRNA and protein expression of ASPN was found to be higher in the endometriosis group than in normal endometrial tissue (*p* = 0.019, n = 12 vs. 12; *p* = 0.0013, n = 10 vs. 10) (Figure 5C–E).

### 3.3. Sensitivity and Specificity of ASPN in Internal and External Datasets

To further clarify the sensitivity and specificity of ASPN for the diagnosis of endometriosis, we evaluated the diagnostic efficacy of ASPN in three datasets. The sensitivity of ASPN in the GSE141549 dataset was 77.3%, and the specificity was 65.2%; the sensitivity was 90% and the specificity was 90% in the 7305 data set. In the clinical case-cohort we collected, the sensitivity of ASPN was 91.7%, and the specificity was 66.7% (Table 3), which suggests that ASPN may be a potential biomarker for the diagnosis of endometriosis.

### 3.4. Correlation of ASPN Expression with Clinical Features in Patients with Endometriosis

To gain insight into the clinical significance of ASPN, we investigated the relationship between ASPN and the clinical features of endometriosis patients. The results showed that ASPN was associated with age (*p* = 0.03), r-AFS stage (*p* = 0.006), and lesion location (*p* < 0.001) (Table 4), which suggests that ASPN is positively correlated with the severity of endometriosis. According to the findings of logistic regression models, elevated ASPN expression is an independent risk factor in the development of endometriosis (Table 5), suggesting that abnormal expression of ASPN is closely related to the development of endometriosis.

### 3.5. Functional Annotation of DEGs and GSEA

To determine the potential function of ASPN and its potential impact on endometriosis, we performed GO and KEGG enrichment analysis of 255 differential genes between high- and low-expression groups. Figure 6 shows that DEGs were mainly enriched in drug metabolism cytochrome P450, extracellular matrix structural constituent, collagen containing extracellular matrix, and extracellular structure organization. To further investigate the possible involvement of ASPN in signaling pathways, we performed a GSEA between high and low ASPN expression phenotypes. GSEA results reveal that the enriched pathways include vascular smooth muscle contraction, cytokine interactions, calcium signaling, and chemokine signaling pathways. The enrichment pathway is considered to be meaningful, according to the cutoff value of *p* < 0.05.

### 3.6. ASPN Is Associated with Immune Infiltration in Endometriosis

To determine the potential function of ASPN and its impact on endometriosis, based on the median of ASPN as the cutoff value, we split our 401 samples into two groups, one with high ASPN expression and one with low ASPN expression. We estimated the degree of infiltration of each sample immune cell subtype using the MCP-counter. T cells, B cells, cytotoxic lymphocytes, NK cells, neutrophils, myeloid dendritic cells, endothelial cells, and fibroblasts are the main immune cells affected by ASPN expression. The box plot of immune cell infiltration fraction showed that T cells, B cells, neutrophils, myeloid dendritic cells, endothelial cells, and fibroblasts were significantly increased in the high expression group; cytotoxic lymphocytes and NK cells were decreased in the high expression group (Figure 7A).

Then, we analyzed the interaction between ASPN expression and the level of immune cell infiltration. Spearman correlation analysis showed a significant positive correlation between ASPN expression and the infiltration abundance of T cells (r = 0.33), B cells (r = 0.22), neutrophils (r = 0.14), endothelial cells (r = 0.56), and fibroblasts (r = 0.54) and a significant negative correlation with cytotoxic lymphocytes and NK cell infiltration fraction abundance (Figure 7B). To further investigate the potential role of ASPN in infiltrating immune cells, we explored the relationship between ASPN and various immune cell markers; as shown in Table 5, the expression of ASPN was positively correlated with the expression levels of CD2, CD3E, CD79A, CD19, CCR7, PECAM1, VWF, and RGS5, but negatively correlated with the expression levels of KIR2DL1, KIR2DL3, KIR2DL4, and KIR3DL3. The findings above imply a relationship between stromal cells and immune cells in the immunological microenvironment and ASPN genes. Aberrant expression of ASPN is associated with the formation of the immune microenvironment and may play a vital role in the onset, advancement, metastasis, and immune response of endometriosis.

To investigate ASPN’s synergistic effects on the induction of immune responses in endometriosis, correlation analysis of ASPN expression levels with immune checkpoint molecules showed that ASPN expression was highly correlated with CTLA4, LAG3, ICOS, CD48, CD27, and CD40 (*p* < 0.05) (Figure 7C). This suggests that ASPN is a potential co-regulator of immune checkpoints in endometriosis.

## 4. Discussion

Endometriosis is a chronic inflammatory disease that depends on estrogen. Patients with endometriosis frequently miss the best chance for treatment due to the lack of sensitive and reliable diagnostic indicators, which has a negative impact on their prognosis [22]; therefore, a timely and appropriate diagnosis of endometriosis is essential to improve the prognosis. In addition, previous research has revealed that endometriosis growth and progression are significantly influenced by changes in immune cell quantity and function [23]. Therefore, exploring the mechanisms of the role of key molecules and immune cells in the pathophysiology of endometriosis will help to elucidate, to some extent, the specific role of the immune system in the pathogenesis of endometriosis, thus providing potential immunotherapeutic targets for future immunotherapy.

In our study, we downloaded relevant transcript data from the GEO database, compared the gene expression profiles of endometriosis tissues and normal samples, screened out genes with high expression differences, and identified ASPN as a possible diagnostic biomarker for endometriosis through machine learning algorithms and external validation. ASPN mRNA expression levels were significantly elevated in patients with endometriosis. In addition, the western blot assay confirmed that the expression of ASPN protein was higher than that of normal tissues. We conducted an ROC curve study to confirm the diagnostic efficacy of ASPN further, and the results indicated that ASPN has a high AUC value, with a sensitivity of 77.3% and a specificity of 65.2%. This suggests that ASPN may be a promising biomarker for differentiating endometriosis from normal controls.

ASPN is a gene found in bone and joint diseases that encodes a small leucine-rich proteoglycan, an important component of extracellular matrix tissue proteins [24]. ASPN has been reported to be involved in the pathophysiological processes of various diseases, including cancer, osteoarthritis, and intervertebral disc disease [25,26,27]. Extracellular ASPN regulates the interaction of TGF-β with its receptor and acts as a negative regulator of TGF-β, promoting bone metastasis in prostate cancer cells. ASPN secreted by chondrocytes binds to type I collagen through its LRR domain, affects collagen fiber production, and is involved in regulating the pathogenesis of osteoarthritis [28,29]. Cytoplasmic ASPN interacts with Smad2/3 to promote its translocation to the nucleus and activates the TGF-β/Smad2/3 signaling pathway to promote EMT and proliferation in various cancer cells [30]. Meanwhile, ASPN is regulated by multiple miRNAs that affect disease regression. ASPN is significantly up-regulated in osteoarthritis, and miR-4303 targets negative regulation of ASPN expression to alleviate inflammation in chondrocytes [31]. miR-101 inhibits colorectal cancer progression through targeted modulation of ASPN. In colorectal cancer, ASPN expression was significantly elevated and correlated with advanced clinicopathological features such as tumor size, disease recurrence, and poor prognosis. Increased gene copy number variation is the main reason for the overexpression of ASPN in colorectal cancer tissues [32].

In this study, we found for the first time that ASPN was highly expressed in endometriosis tissues and was also correlated with patients’ age (*p* = 0.03), r-AFS stage (*p* = 0.006), and lesion location (*p* < 0.001), suggesting a positive correlation between ASPN and the severity of endometriosis. The results of constructing univariate and multivariate logistic regression models showed that high expression of ASPN was an independent risk factor for the development of endometriosis, suggesting that abnormal expression of ASPN is closely related to the development of endometriosis.

Exploring the molecular mechanisms associated with ASPN in endometriosis will help further investigate new targeted therapeutic approaches. We divided patients into high and low ASPN subgroups based on the median value of ASPN expression to establish a functional network based on differential genes. GO and KEGG pathway analysis revealed that these differential genes associated with ASPN expression are involved in a wide range of biological processes, including structural components of the cellular matrix, the collagen-containing cellular matrix, extracellular structural organization, and drug metabolism-cytochrome P450-related signaling pathways [33]. Through GSEA, overexpression of ASPN was found to be associated with multiple signaling processes, such as vascular smooth muscle contraction, cytokine interactions, calcium signaling pathways, and chemokine signaling pathways [34,35]. These signaling processes and biological pathways are associated with the development of endometriosis. However, further experiments are needed to validate the mechanisms of these pathways regulated by ASPN in endometriosis.

Immune cell infiltration and immune dysfunction in the immune microenvironment of endometriosis are involved in the development of the disease; however, the relationship between immune infiltration and ASPN expression has not been reported. Using the MCP counter technique to assess the amount of immune cell infiltration in endometriosis, we discovered increased infiltration of T cells, B cells, and fibroblasts and decreased infiltration of cytotoxic lymphocytes and natural killer cells. T cell subsets Th1, Th2, Th17, and regulatory T cells promote disease progression by secreting a variety of cytokines that regulate the function of other immune cells, facilitating ectopic implantation and proliferation of endometrial cells, and increasing angiogenesis [36,37]. Increased numbers and activity of B lymphocytes play an important role in the pathogenesis of endometriosis by regulating immune cell activity and producing large amounts of anti-endometrial antibodies [38]. Meanwhile, abnormal expression of NK cell receptors and reduced cytotoxicity of NK cells, the first line of defense of the body’s immune system, is involved in the development of endometriosis [39]. Our findings are consistent with previous reports.

In addition, T cells, B cells, and fibroblast infiltration levels were strongly correlated with the transcript levels of the ASPN gene, while cytotoxic lymphocytes and NK cells were negatively correlated. Subsequent analysis revealed that the expression of ASPN was significantly correlated with the expression of immune marker genes. In terms of immune checkpoint molecules, ASPN expression was closely associated with CTLA4, LAG3, CD27, CD40, and ICOS. The above results suggest a potential relationship between ASPN and immune cell infiltration in endometriosis and that ASPN may be a potential target for immunotherapy.

Our study is based on secondary mining of public database data. It lacks corresponding clinical information, and prospective large-scale clinical investigations are required to look into the connection between ASPN and endometriosis in more detail. Secondly, the study of ASPN is mainly carried out through bioinformatics analysis. Further in vitro and in vitro experiments are needed to elucidate the detailed mechanism of ASPN’s involvement in the pathogenesis of endometriosis.

In the present study, for the first time, we discovered that ASPN is strongly expressed in endometriosis and is related to immune cell infiltration, particularly T cells, B cells, and NK cells. ASPN is considered a new potential biomarker involved in the pathogenesis of endometriosis. This study contributes to further understanding of the mechanisms underlying the development of endometriosis and provides new ideas for discovering drug targets for endometriosis.

## Figures and Tables

**Figure 1 genes-13-01352-f001:**
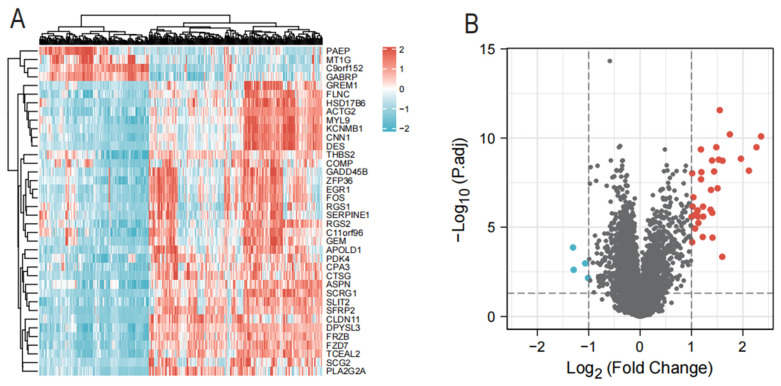
Differentially expressed genes (DEGs) of GSE141549. (**A**) Heat maps of 38 DEGs found in endometriosis tissue versus control samples. (**B**) DEG volcano plot of endometriosis tissue and control samples.

**Figure 2 genes-13-01352-f002:**
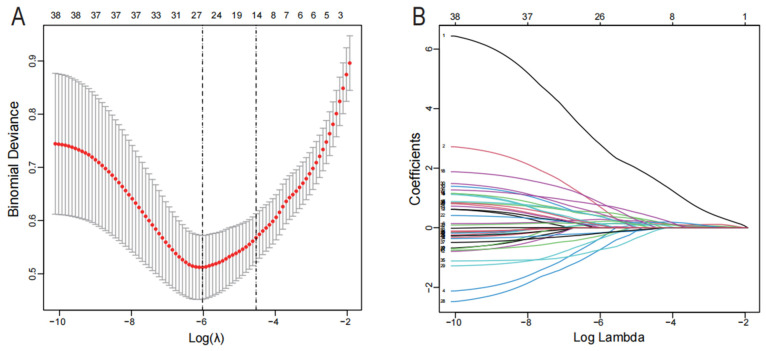
Screening of diagnostic markers via the Lasso regression. (**A**) Lasso coefficient distribution misclassification error. (**B**) Different colors correspond to various genes in the distribution of Lasso coefficients for 38 related genes.

**Figure 3 genes-13-01352-f003:**
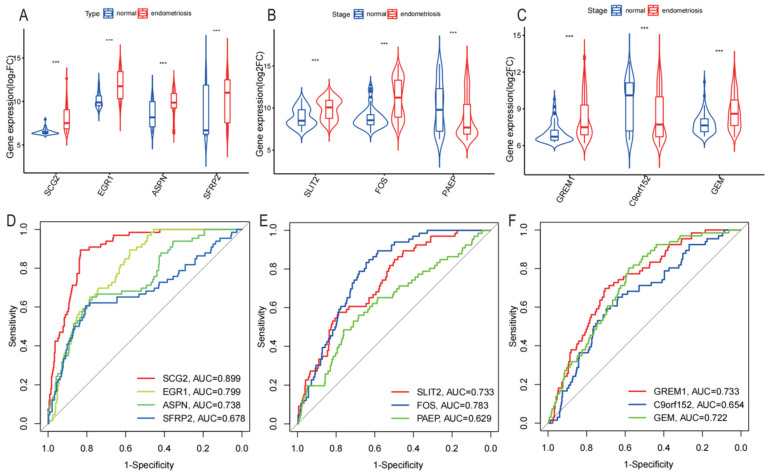
Validation of diagnostic markers by internal datasets. (**A**–**C**) Validation of the expression of 10 potential biomarkers in the 141549 dataset. (**D**–**F**) Validation of the diagnostic effectiveness of 10 biomarkers in the 141549 dataset (*** *p* < 0.001).

**Figure 4 genes-13-01352-f004:**
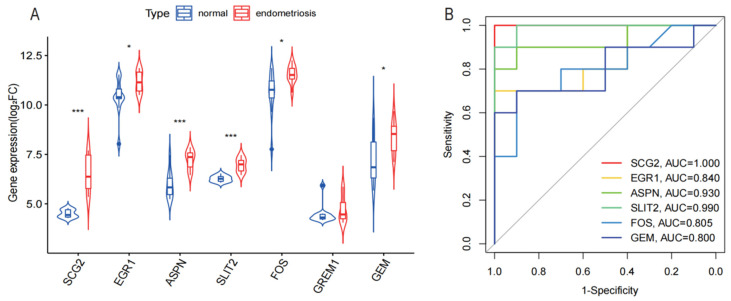
Validation of diagnostic markers by external dataset. (**A**) Validation of the expression of seven potential biomarkers in the 7305 dataset. (**B**) Verification of the diagnostic effectiveness of six biomarkers in the 7305 dataset (* *p* < 0.05, *** *p* < 0.001).

**Figure 5 genes-13-01352-f005:**
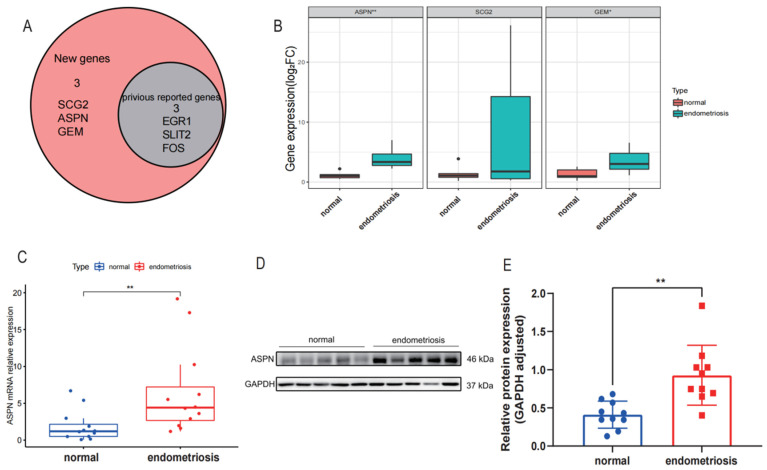
Validation of the expression of important biomarkers and screening for new biomarkers. (**A**) New biomarker for endometriosis. (**B**) The degree of ASPN, SCG2, and GEM mRNA expression in endometriosis and control samples. (**C**) Normal samples and Endometriosis ASPN mRNA expression levels. *p* = 0.0045 for mean±SD. (**D**,**E**) Endometriosis and control sample ASPN protein expression levels. *p* = 0.0013 for mean ± SD (* *p* < 0.05, ** *p* < 0.01).

**Figure 6 genes-13-01352-f006:**
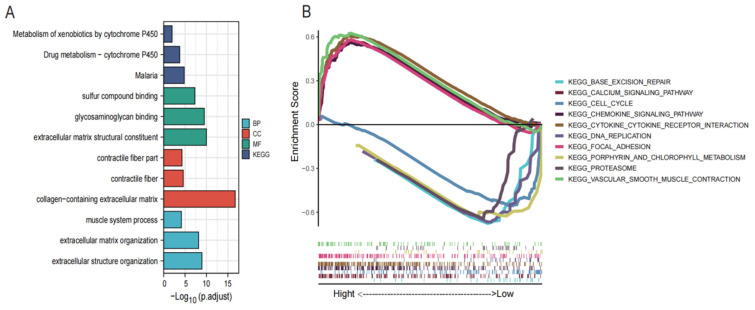
Functional annotations of DEGs. (**A**) GO and KEGG functional enrichment of DEGs. (**B**) GSEA for significantly enriched pathways.

**Figure 7 genes-13-01352-f007:**
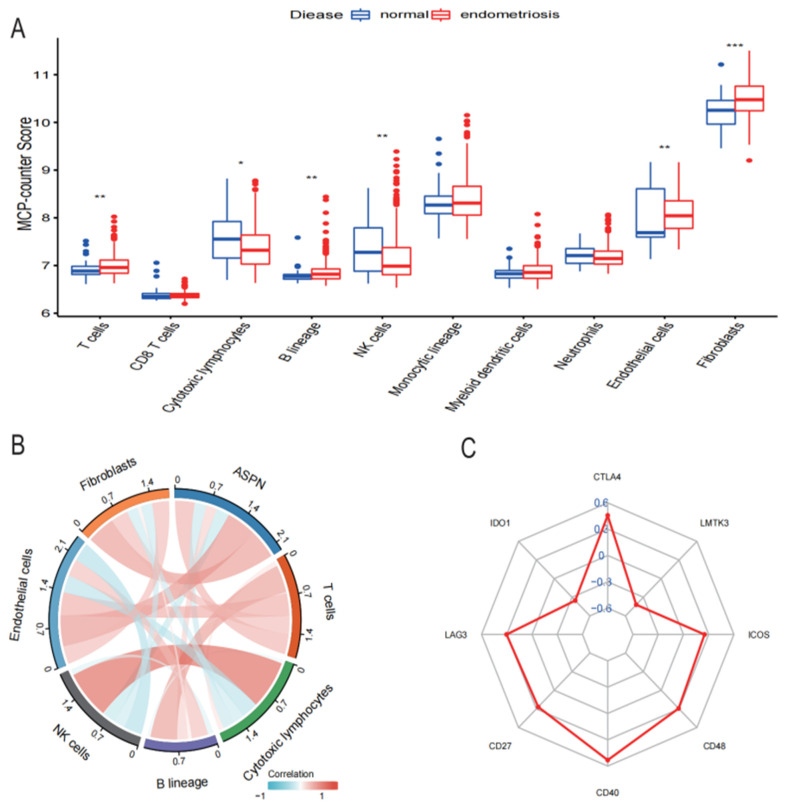
Distribution of immune cell infiltration and correlation analysis. (**A**) The composition of immune cells in endometriosis tissues and normal control tissues. (**B**) Analysis of the correlation between ASPN and immune cells invading endometriosis. (**C**) The relationship between the immune checkpoint genes and ASPN (* *p* < 0.05, ** *p* < 0.01, *** *p* < 0.001).

**Table 1 genes-13-01352-t001:** Clinical characteristics of the GSE141549 dataset.

Characteristics		Number of Cases
Age (years)	≤35	241
	>35	160
Stage		
	I–II	85
	III–IV	250
	Healthy	66
Cycle Phase		
	menstruation	20
	proliferative	60
	secretory	97
	medication	175
	unknown	49
Site		
	endometrium	146
	endometriosis	166
	DIE	89

**Table 2 genes-13-01352-t002:** Generalized linear model screening results.

Gene Names	HR (95% CI)	*p*-Value
SCG2	7.89 (4.80–11.74)	<0.001
EGR1	3.70 (2.10–5.82)	<0.001
ASPN	1.58 (1.04–2.32)	0.038
SLIT2	3.07 (1.80–4.76)	<0.001
GEM	0.17 (0.03–0.63)	0.008
SFRP2	0.34 (0.15–0.65)	0.001
FOS	0.24 (0.08–0.56)	0.001
GREM1	0.32 (0.09–0.90)	0.035
C9orf152	2.01 (1.31–2.94)	0.003
PAEP	0.69 (0.51–0.89)	0.007

**Table 3 genes-13-01352-t003:** Sensitivity and specificity of ASPN in internal and external datasets.

Dataset	Number	AUC (%)	Yoden Index	Sensitivity (%)	Specificity (%)
GSE141549	401	73.8	0.43	77.3	65.2
GSE7305	20	93.0	0.80	90.0	90.0
Clinical Data	24	83.3	0.58	91.7	66.7

**Table 4 genes-13-01352-t004:** Clinical characteristics of endometriosis patients with low and high expression of ASPN.

		ASPN		
Clinical Parameters	n (%)	High Group (%)	Low Group (%)	*p*-Value
Age (years)		32.24 ± 6.694	33.60 ± 7.387	0.030
Stage				
I–II	85 (21.2)	45 (11.2)	40 (10.0)	0.006
III–IV	250 (62.3)	134 (33.4)	116 (28.9)	
Healthy	66 (16.5)	21 (5.2)	45 (11.2)	
Cycle Phase				
menstruation	60 (15.0)	29 (7.3)	31 (7.7)	0.227
proliferative	97 (24.2)	42 (10.5)	55 (13.7)	
secretory	20 (5.0)	14 (3.5)	6 (1.5)	
medication	175 (43.6)	92 (22.9)	83 (20.7)	
unknown	49 (12.2)	23 (5.7)	26 (6.5)	
Site				
Endometrium	146 (36.4)	22 (5.5)	124 (30.9)	<0.001
Peritoneum	27 (6.7)	4 (1.0)	23 (5.7)	
Endometriosis	139 (34.7)	100 (24.9)	39 (9.7)	
DIE	89 (22.2)	74 (18.5)	15 (3.7)	

**Table 5 genes-13-01352-t005:** Logistic regression to determine the effect of different clinicopathological factors on the expression of ASPN gene and clinical prognosis in endometriosis.

	Univariate-Logistic		Multivariate-Logistic	
Clinical Parameters	HR (95% CI)	*p*-Value	HR (95% CI)	*p*-Value
Age (years)	0.842 (0.802–0.884)	<0.001	0.848 (0.805–0.894)	<0.001
Cycle Phase	1.077 (0.884–1.313)	0.462	/	/
Site	1.895 (1.472–2.438)	<0.001	1.256 (0.867–1.819)	0.229
ASPN	1.870 (1.544–2.266)	<0.001	1.574 (1.199–2.068)	0.001

## Data Availability

Publicly available datasets were analyzed in this study. This data can be found here: https://www.ncbi.nlm.nih.gov/geo/ (accessed on 20 December 2021); GSE141549, GSE7305.

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
