# Peer review of "ASPN Is a Potential Biomarker and Associated with Immune Infiltration in Endometriosis"

_genes, 2022, doi:10.3390/genes13081352_

Round 1

Reviewer 1 Report

In this study, Wang et al performed an interesting study to identify the markers for immune infiltration in endometriosis. I speculate that the study would generate a lot of interest and may generate significant therapeutic values in future. Overall, I think this is a decent study that is thoroughly designed and thoughtfully executed but I don’t think the manuscript is written well enough to explain how the authors have strategized the study and also the full impact of the findings. For example,

(1) In the abstract, the authors say that: “We obtained the GSE141549 15 and GSE7305 datasets from the GEO database. Differentially expressed genes (DEGs) were identified in the GSE141549 dataset.”

This is the first time authors are introducing their workflow but surprisingly, they don’t say what these datasets are. The readers don’t know about these datasets. So they need to explain these datasets in the beginning so that people can understand their strategy. This is a very basic thing while writing a manuscript.

(2) Similarly, in the last paragraph of introduction the authors say that “Download the endometriosis data in the GEO database and use some bioinformatics algorithms and tools to screen for possible diagnostic markers of endometriosis.”

This sentence does not make any sense grammatically, and on top, it is completely vague to visualize what the authors are doing in this study. In the introduction section of the study, the authors need to do a very good job to explain their strategy of the whole study based on which the readers are going to look into the results. Without that, I am afraid to say, it would be hard for anyone to focus on the findings explained later.

Given that these are very fundamental issues of writing a manuscript, I believe that the authors need to majorly think about re-write/edit these sections before this can be published. Without this, even with the good quality of the data, the main findings of the study can not be highlighted.

Author Response

Dear Reviewers

Thank you very much for your review of my manuscript and for the valuable corrections you provided. We have made the following changes to the manuscript based on your comments.

To clarify the diseases to which the dataset belongs, We change “We obtained the GSE141549 15 and GSE7305 datasets from the GEO database ” to “We obtained the GSE141549 and GSE7305 datasets containing endometriosis and normal endometrial samples from the Gene Expression Omnibus database (GEO).” in the abstract.

In order to fully explain the whole strategy of the study and make the results easily understandable to the reader, We change “Download the endometriosis data in the GEO database and use some bioinformatics algorithms and tools to screen for possible diagnostic markers of endometriosis” to “In our study, the GSE141549 dataset was downloaded from GEO to identify DEGs between endometriosis and controls. Lasso regression and generalized linear model algorithms are used to screen diagnostic biomarkers for endometriosis. We confirmed the diagnostic efficacy of biomarkers in the GSE7305 dataset. Further validation of the expression levels of the biomarkers was carried out using qRT-PCR and Western-blot tests. The study found that ASPN may be a new biomarker for endometriosis. Univariate and multifactorial logistic regression models were further constructed to analyze the correlation between ASPN expression and clinical characteristics of patients with endometriosis. Finally, the immune cell composition between endometriosis and normal tissue was quantified using the MCP-counter. By studying the relationship between ASPN, immune infiltration, and immune checkpoint genes, we can better understand the molecular immune mechanisms involved in the pathogenesis of endometriosis.” in the introduction.

Reviewer 2 Report

The manuscript entitled “ASPN is a potential biomarker and associated with immune infiltration in endometriosis” is very interesting. The manuscript is well written. However, I highly recommend the manuscript to go through minor revision and address the following questions before publication:

1.      Several spacing errors found, such as in L15, L25, L139. Recheck the manuscript carefully.

2.      The “p” of p value should be in italics.

3.      Provide the catalog number of the primary antibodies used in the study.

4.      Change Figure 3 in tubular form.

5.      In figure 4 and 5, is the gene expression represented in fold change or in percentage?

6.      I could not understand what authors mean to say with “Genes that have been previously reported in the literature to be associated with the development of endometriosis (SCG2, ASPN, GEM) were excluded, and three new biomarkers (SCG2, ASPN, GEM) were screened, and the results are shown in Figure 6A”. I think there is a typological error. Authors might want to say (EGR1, SLIT2, and FOS) are the genes which was previously reported, and in this study, they found three new genes as SCG2, ASPN, GEM, that could be used as biomarkers which are associated with the development of endometriosis!

7.      I suggest to be consistent with the sign to denote p-value, it could be either the asterisk or the value itself.

8.      In figure 6D, does lane represent an individual person participating in the study?

Author Response

Dear Reviewers

Thank you very much for the valuable corrections you have provided to me. By revising the details of the article to make it more readable and understandable, we have made the following changes for this purpose:

  1. We rechecked the manuscript carefully to remove the spacing error;
  2. The P used in the text was replaced by "p".
  3. Catalog numbers of antibodies were added;
  4. Figure 3 was changed to a tabular format; for Figures 4 and 5, gene expression is expressed as a fold change;
  5. We change “Genes that have been previously reported in the literature to be associated with the development of endometriosis (SCG2, ASPN, GEM) were excluded, and three new biomarkers (SCG2, ASPN, GEM) were screened, and the results are shown in Figure 5A. ” to “Three novel biomarkers (SCG2, ASPN, and GEM) were screened, and the findings are displayed in Figure 5A. The genes( EGR1, SLIT2, and FOS) were rejected because they have been previously correlated with the development of endometriosis in the literature.” in the result.
  6. Pvalues are uniformly normalized with an asterisk throughout the text;
  7. The proteins in Figure 6D were extracted from samples of tissues, which were obtained from human endometriosis tissue and normal endometrial tissue. Each band represents a case sample.

Reviewer 3 Report

The manuscript from Wang and Sun provides interesting study about ASPN as a biomarker for endometriosis disease. Claims in the manuscript are largely supported by experimental evidence and the association of ASPN expression with immune cell infiltration during endometriosis is interesting. Before publication, manuscript should be benefited by addressing the following comments.

Major comments:
1. Introduction needs reorganization. For instance, at present, introduction to the disease endometriosis and its pathological changes is not sufficient. Details are also required about current diagnostic methods like ultrasound, magnetic resonance imaging (MRI) or laparoscopy and their limitations to justify the need for new biomarkers.
2. In the abstract, the authors say that (line 14) “We aimed to find potential biomarkers of endometriosis and evaluated the significance of immune cell infiltration in pathogenesis”
I don’t think data related to the significance of immune cell infiltration is provided in this manuscript. Therefore, if authors wish to keep the above statement, then they need to provide supporting data.
3. In the result section, brief details are required about Lasso regression & linear model with respective citation (Line 184-187).
4. Line 207-209 needs correction with respect to know versus new biomarker. At present, both are same
“Genes that have been previously reported in the literature to be associated with the development of endometriosis (SCG2, ASPN, GEM) were excluded, and three new biomarkers (SCG2, ASPN, GEM) were screened, and the results are shown in Figure 6A”
5. One of the characteristic features of a disease biomarker is that it should be specific for that disease. Thus, in this study, it is not clear if ASPN is specific to endometriosis or a generally activated gene in disease related to the uterus (like Uterine Fibroids and Polycystic Ovary Syndrome (PCOS)) or one which results in tissue inflammation. In case, authors are not able to show specificity of ASPN with endometriosis, then they should write it as the limitation of their study.

Minor:
1. In figure 6, the sequence of normal versus endometriosis samples data is not uniform. For instance, in Fig 6B and Fig 6E the sequence is opposite, which creates confusion.
2. When using any abbreviated form for the first time, giving full form is helpful for the reader. MCP-counter method is used in abstract without giving full form of MCP. This needs to be corrected for all abbreviations.
3. More information is required for Figure 2B and 3B legends, especially with respect to the color code used in the graph.

Author Response

Dear Reviewers

Thank you very much for your valuable comments on our manuscript, and we have revised it accordingly:

We accepted the valuable comments of the reviewers and reorganized the introduction with the following additions. “In recent years, treatment techniques related to endometriosis have been advancing. Still, the effectiveness of existing standard treatments is limited due to the aggressive nature of endometriosis and the high postoperative recurrence rate. Laparoscopy is the gold standard for the diagnosis of endometriosis, but the invasive nature of the technique limits its widespread clinical use[3]. Ultrasound and magnetic resonance imaging are commonly used to diagnose and evaluate endometriosis conditions. Still, the sensitivity of imaging varies depending on the location of the endometriosis lesion, while the detection rate of imaging for early endometriosis lesions is extremely low[4, 5]. Many biochemical markers are widely used in clinical diagnosis, including CA125, HE4, etc. However, these methods are insufficient for the early diagnosis of endometriosis due to sensitivity and specificity limitations. Therefore, it is essential to investigate more potential molecular targets for their diagnosis, treatment, and prognostic assessment.”

We change “We aimed to find potential biomarkers of endometriosis and evaluated the significance of immune cell infiltration in pathogenesis.” to “We aimed to find potential biomarkers of endometriosis and to explore the relationship between ASPN and the immune microenvironment of endometriosis.” in the abstract.

Brief details of the lasso regression and linear models are added in the results section, and relevant references are inserted.

We change “Genes that have been previously reported in the literature to be associated with the development of endometriosis (SCG2, ASPN, GEM) were excluded, and three new biomarkers (SCG2, ASPN, GEM) were screened, and the results are shown in Figure 5A. ” to “

Three novel biomarkers (SCG2, ASPN, and GEM) were screened, and the findings are displayed in Figure 5A. The genes( EGR1, SLIT2, and FOS) were rejected because they have been previously correlated with the development of endometriosis in the literature.” in the result.

The section "Sensitivity and specificity of ASPN in internal and external datasets" was added to the results to further elucidate the possibility that ASPN may be a specific gene for endometriosis.

The order of the normal and endometriosis sample data in Figure 6 was adjusted and aligned.

Abbreviations throughout the text were checked and revised, and highlighted in a special color font.

Figure 3 was changed to a tabular form based on reviewer 2's comments, and the notes to Figure 2B were further supplemented as follows:”FIGURE 2 |Screening of diagnostic markers via the Lasso regression. (A) Lasso coefficient distribution misclassification error. (B) Different colors correspond to various genes in the distribution of Lasso coefficients for 38 related genes.”

Round 2

Reviewer 1 Report

The author's revision made this manuscript significantly improved. The abstract and introduction now nicely lead into the data/results without any confusion. I agree that this would be a nice addition in MDPI-Genes. Congratulations to the team. 

Reviewer 3 Report

Authors addressed all my comments. Best wishes!!